# Measuring health-related quality of life in the general population and Roma communities in Romania: study protocol for two cross-sectional studies

Elena Olariu,[1] Marian Sorin Paveliu,[2] Eugen Baican,[3] Yemi Oluboyede,[1] Luke Vale,[1] Ileana Gabriela Niculescu-Aron[4]

[1]Health Economics Group, Institute of Health and Society, Newcastle University, Newcastle upon Tyne, UK
[2]Department of Pharmacology and Pharmacoeconomics, Universitatea Titu Maiorescu Facultatea de Medicina, Bucuresti, Romania
[3]Department of Social Work, Universitatea Babes-Bolyai, Cluj-Napoca, Romania
[4]Department of Statistics and Econometrics, Faculty of Economic Cybernetics, Statistics and Informatics, Academia de Studii Economice, Bucuresti, Romania

**Correspondence to**
Dr Marian Sorin Paveliu; sorinpaveliu@yahoo.com

## ABSTRACT

**Introduction** The importance of health-related quality of life (HRQoL) is increasing and many healthcare authorities recommend the use of measures that account for both mortality and morbidity. This study will determine, for the first time in Romania, value sets for EuroQoL-five-dimensions-3-level (EQ-5D-3L) and EQ-5D-5L questionnaires and their population norms (study 1). It will also compare the HRQoL (measured with EQ-5D-5L) of Roma communities in Romania with that of the general population (study 2).

**Methods and analysis** Cross-sectional studies of face-to-face interviews conducted in representative samples of the Romanian general population and Romanian Roma communities. 1614 non-institutionalised adults older than 18 years will be interviewed using a computer-assisted interview for study 1. Participants will complete EQ-5D-3L and 5L, 13 composite time trade-off tasks (cTTO), 7 discrete choice experiment questions (DCE) and sociodemographic questions. For study 2, 606 non-institutionalised self-identified Roma people older than 18 years will be interviewed using a pencil-and-paper interview. Participants will complete EQ-5D-5L and the same sociodemographic questions as for study 1. The 3L value set will be estimated using econometric models and the cTTO data. cTTO and DCE data will be used for the 5L value set. Population norms will be reported by age and gender. The ORs for reporting different levels of problems and the most common health states in the population will be estimated. For study 2, t-tests and analysis of variance will be used to explore differences between groups in HRQoL and for each EQ-5D.

**Ethics and dissemination** Ethics approval was given by the National Bioethics Committee of Medicines and Medical Devices Romania and Newcastle University's Research Ethics Committee. Results will be published in peer-reviewed journals, presented at scientific conferences and on the project's website. The EQ-5D-5L anonymised datasets will be deposited in a centralised repository. Two public workshops with local authorities, physicians and patients' associations will be held.

## INTRODUCTION

Health-related quality of life (HRQoL) is a multidimensional concept that includes

### Strengths and limitations of this study

► First studies to determine the value sets and population norms for both EuroQoL-five-dimensions-3-level (EQ-5D-3L) and 5L for Romania.
► First studies to determine health-related quality of life (as measured with EQ-5D-5L) of Roma population from Romania.
► The representative samples include only persons who reside in the country at the time of the data collection. Romania has one of the highest migration rates in Europe: 17% of the total population of Romania (3.3 million of 19.6 million) are working abroad, many of them are Roma ethnicity.

physical, psychological, functional and social aspects[1] regarding a person's perception of the impact of health status on quality of life.[2]

In the last 20 years, the importance of HRQoL in health research has increased with many healthcare authorities recommending the use of indicators, such as quality-adjusted life-years (QALYs), rather than relying on just indicators of mortality or morbidity alone.[3] QALYs are the mainstay of economic evaluations that help decision-makers to determine how resources can be used to give the greatest benefits.[4] QALYs combine information on the length of life with information on quality of life, expressed as utilities. Several tools are available to measure individuals' health status and to attach utility values to these health states, such as the EuroQoL-five-dimension (EQ-5D),[5 6] Health Utilities Index,[7 8] Short Form 6-dimension[9] and so on.

The EQ-5D is one of the most commonly used questionnaires to elicit health states utilities, having both a youth[10] and an adult version. For general populations of adults two versions of the questionnaire have been developed, 3L and 5L, with the latter offering a more detailed description of health status.[6]

In order to use the EQ-5D to estimate QALYs, a value set to reflect people's preferences for different health states is needed. At present, no value sets have ever been developed in Romania for either the 3L or 5L versions of the questionnaire. Researchers and decision-makers have always used the value set from another jurisdiction, most commonly the value sets in the UK.[11–14] Using value sets from other countries might introduce bias as people's preferences can be culturally sensitive, differing from one country to another.[15] Therefore, local reimbursement and health technology assessment (HTA) decisions should be based on the data applicable to the local context to avoid incorrect and inefficient decisions about the allocation of resources.[16]

Besides their use in economic evaluations, tools, such as the EQ-5D, can be used to identify health problems in the community. The quality of life of patients or vulnerable groups can be compared with that of an average person from the population of interest with the help of population norms. This can help identify health inequalities between different groups. This, in turn, may help identify where healthcare and public health interventions are needed to reduce inequalities. This is of particular relevance to Romania where health inequalities have increased sharply in the past few years between regions and groups.[17] One particularly vulnerable group is represented by the Roma communities whose health status is consistently reported as being poorer than that of the general population.[18] Up to this date, the HRQoL of either the general population or Roma communities has never been determined with the EQ-5D questionnaire and no population norms exist in Romania for either of the two versions of the questionnaire.

The aim of this study is to determine the value sets and population norms for the EQ-5D questionnaires and to compare HRQoL (measured with EQ-5D-5L) between the general population and Roma communities.

## METHODS AND ANALYSIS
### Study design
A cross-sectional study design consisting of face-to-face interviews conducted in two national representative samples (general population and Roma communities) recruited from all regions of Romania will be conducted from November 2018 to April 2019.

### Aim and objectives
Our survey aims to provide HRQoL data to support HTA and reimbursement decisions in Romania and to support the development of interventions that are more responsive to the needs of local communities in Romania. Our specific objectives are:
1. To develop a value set for EQ-5D-3L and 5L based on societal preferences in Romania (study 1, an Omnibus survey).
2. To determine the population norms of the Romanian version of EQ-5D-3L and 5L (study 1, an Omnibus survey).

3. To describe differences in HRQoL between Roma communities and the majority of the population according to sociodemographic characteristics, socioeconomic status and ethnicity (study 2).

### Study population
#### Target population
##### *Study 1*
Non-institutionalised adults older than 18 years who reside in the country at the time of the data collection.

##### *Study 2*
Non-institutionalised self-identified Roma adults older than 18 years who reside in the country at the time of the data collection.

#### Sampling frame
##### *Study 1 and study 2*
The sampling frame consists of all polling stations in Romania (18 626 polling stations for 3181 settlements) that are publicly available on the website of the Romanian Permanent Electoral Authority.

#### Sample design
##### *Study 1*
The sample was selected using a three-stage probability sampling procedure stratified by region and settlement size. Romania's territory was divided into strata using two criteria built according to the recommendations of the National Institute of Statistics:
1. Regional divisions (development regions):
   A. North–East: Bacău, Botoşani, Iaşi, Neamţ, Suceava, Vaslui.
   B. South–East: Brăila, Buzău, Constanţa, Galaţi, Tulcea, Vrancea.
   C. South: Argeş, Călăraşi, Dâmboviţa, Giurgiu, Ialomiţa, Prahova, Teleorman.
   D. South–West: Dolj, Gorj, Mehedinţi, Olt, Vâlcea.
   E. West: Arad, Caraş-Severin, Hunedoara, Timiş.
   F. North–West: Bihor, Bistriţa-Năsăud, Cluj, Maramureş, Satu Mare, Sălaj.
   G. Centre: Alba, Braşov, Covasna, Harghita, Mureş, Sibiu.
   H. Bucharest: Ilfov, Bucharest.
2. Size of settlement:
   A. Bucharest (more than 1 million inhabitants).
   B. Cities with more than 160 000 inhabitants, but up to 1 million inhabitants (≤).
   C. Cities with 50 000–160 000 inhabitants.
   D. Cities with less than 50 000 inhabitants.
   E. Rural settlements.

The strata were obtained by crossing the categories of the two criteria described above: a total of 19 strata were obtained (see table 1). The first three categories of the 'size of settlement' criteria were not crossed with the criteria 'regional divisions' given that the number of settlements in the resulting strata would have been too small.

**Table 1** Number of sampled primary units by development region and settlement size

| Strata no | Strata label | No of settlements (primary units) in the population | No of settlements (primary units) in the sample |
|---|---|---|---|
| 1 | Cities >1 mil inh | 1 | 1 |
| 2 | 160 000 inh. ≤cities < 1 mil inh | 11 | 4 |
| 3 | 50 000 inh. ≤cities <160 000 inh. | 29 | 4 |
| 4 | South: cities <50 000 inh. | 43 | 1 |
| 5 | South-East: cities <50 000 inh | 35 | 1 |
| 6 | West: cities <50 000 inh | 37 | 1 |
| 7 | Centre: cities <50 000 inh | 52 | 1 |
| 8 | North-West: cities <50 000 inh | 37 | 1 |
| 9 | North-East: cities <50 000 inh | 38 | 1 |
| 10 | South-East: cities <50 000 inh | 29 | 1 |
| 11 | Bucharest-Ilfov: cities <50 000 inh | 8 | 1 |
| 12 | South-East: rural settlements | 519 | 2 |
| 13 | South: rural settlements | 408 | 2 |
| 14 | West: rural settlements | 281 | 2 |
| 15 | Centre: rural settlements | 357 | 2 |
| 16 | North-West: rural settlements | 403 | 2 |
| 17 | North-East: rural settlements | 506 | 2 |
| 18 | South-East: rural settlements | 355 | 2 |
| 19 | Bucharest-Ilfov: rural settlements | 32 | 1 |
| | **Total** | **3181** | **32** |

inh, inhabitants; mil, million.

Within each resulting stratum, settlements were selected using a simple random sampling procedure without replacement. A total of 32 settlements were selected (see figure 1): one settlement from the first bracket of the second criteria, eight from the second and third brackets of the second criteria, eight from the fourth bracket of the second criteria (one city with less than 50 000 inhabitants from each development region) and 15 from the last bracket (two rural settlements for the first seven development regions and one rural settlement for Bucharest). The sample size for each stratum was determined based on disproportionate allocation to strata to ensure that at least 30 respondents would be selected for each settlement.

The second stage consists of selecting the polling stations that will be used as a starting point for the 'random route' selection of participants. Polling stations were selected using a simple random sampling procedure without replacement for all settlements selected in the previous stage. The number of polling stations was determined so that a minimum of 20 people could be assigned to each polling station selected from a city and a minimum of 15 people to each polling station selected from a rural settlement. Hence, three polling stations were selected for cities with more than 50 000 inhabitants and two polling stations for cities with less than 50 000 inhabitants and rural settlements. Given that rural settlements can consist of one or more villages, the two polling stations were selected, if needed, from two different villages. In case of Bucharest, two polling sections were selected for each of the six sectors of the city. In total 82 polling stations were selected.

The third stage consists of selecting households and individuals from each settlement and 'voting district'. This will be done using a random route sampling method.[19] Interviewers will perform a random walk per 'voting district'. The interviewer will begin his/her route on the street whose name comes first in alphabetical order from the list of streets that make up the voting district. Once the street has been identified, he/she will start with the household located at the lowest house number on the street. To select the next household, the interviewer will walk on the same side of the street and select the third household in the ascending order.

In each household, the respondent will be selected based on 'birthday rule': the adult whose birthday comes next from the interviewer's visit will be interviewed. In the case of household non-response, the interviewer will visit the household two more times on different days and at different hours. If no contact is established after three calls, then the household is replaced with a new one as per the procedure described above (third household selected).

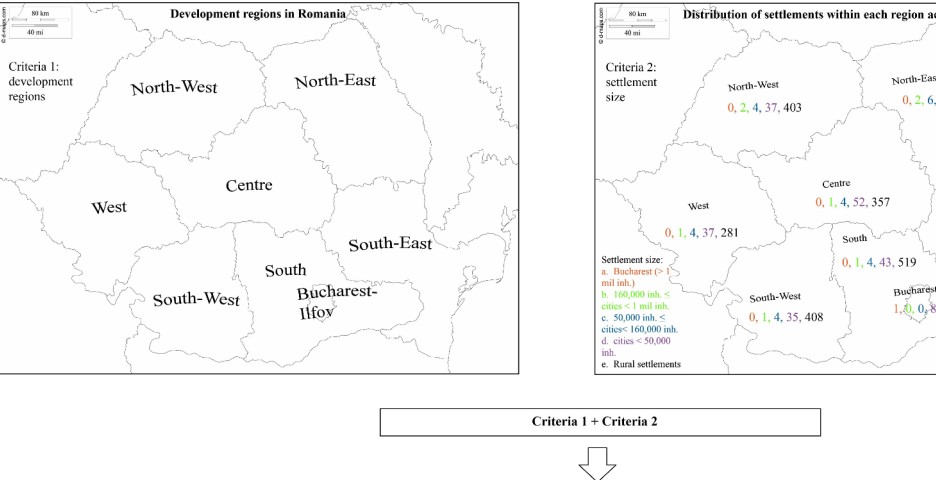

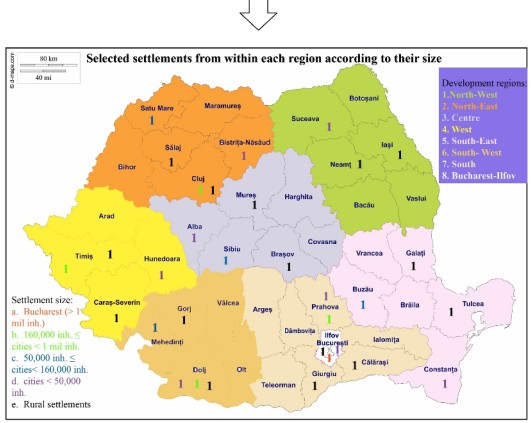

Original maps downloaded from https://d-maps.com/carte.php?num_car=5805&lang=en and
https://d-maps.com/carte.php?num_car=25497&lang=en. Edited by EO.

**Figure 1** Geographical distribution of selected settlements according to development region and size .

### Study 2

In order to increase the comparability of the two samples (general population and Roma communities), study 2 used the same sampling design as study 1. Roma participants were selected from the same settlements selected for study 1. If, based on 2011 census data, no Roma communities were reported to be living in the selected settlements, these were replaced with other settlements from the same strata with higher Roma representation. Disproportionate allocation to strata was used to select the number of respondents from each stratum.

Households and individuals from each settlement will be selected either based on the random route technique ('birthday rule') in those settlements that match the ones selected for study 1 and on a snowball technique for the rest. In case of non-response, the interviewer will make two more attempts on different days and at different hours. If no contact is established after three calls, then the household is replaced with a new one.

### Sample size calculation
### Study 1

The minimum sample size needed to develop a value set for EQ-5D-3L is 300 people[20] and for EQ-5D-5L is 1000 people.[21] Normative data for any outcome instrument need to be obtained from a sample that is an accurate representation of the general population.[22] The sample size needed to select a national representative sample of non-institutionalised Romanian adults aged 18 and older was estimated to be 1613 people, with a maximum error of ±3% at a confidence level of 95%. Due to the complex nature of the survey, the sample size required for a simple random sample was adjusted by the design effect (deff)[23] and by a correction factor to account for the fact that only one person per household was selected (deff(p)). Given that neither the EQ-5D-3L nor 5L has been used before to measure HRQoL in Romania, self-reported health, whereby respondents are asked to classify their current health status, was used as a proxy for HRQoL.

$$N = \frac{t^2 * p\,(1-p)}{\Delta^2} * deff * deff\,(p) = 896*1.5*1.2 = 1,613$$

t=1.96, given a 95% CI; Δ=0.03; p=0.7 (70% of Romanians considered having good health status according to the 2015 National Survey on Quality of Life[24]; deff=1.5[25]; deff (p)=1.2.

The expected non-response rate was set at 10% leading to a final sample size of 1794 people.

### Study 2

Sample size calculations for study 2 were estimated using the same formula described above leading to a sample size of 606 people, with a maximum error of ±5% at a confidence level of 95%. The sample size was similarly

inflated to cover a 10% non-response rate leading to a target sample size of 673 people.

## Pilot study
### Study 1
The study's brochure, informed consent forms and the software needed to collect the data were pretested with approximately 10 volunteers for wording, clarity, order of questions and potential technical problems. Additionally, the local team of investigators performed approximately 30 interviews in Bucharest to gain experience with performing valuation task interviews.

All interviewers attended a 2-day and half face-to-face training in October 2018. Prior to the training, interviewers were asked to read the interviewer's guideline and to familiarise themselves with the data collection tools prior to the face-to-face training. At the end of the training, all interviewers performed approximately five interviews in Bucharest with a convenience sample of respondents selected from hospitals and schools. The local team of investigators assessed the quality of these pilot interviews and provided personalised feedback to each interviewer.

### Study 2
The study's brochure and informed consent forms were pretested with a group of five volunteers from Dâmbovița county and four health mediators. The survey company also pretested the contents of the survey for wording, order of questions and timings with approximately 10 volunteers from Cluj, a city in Romania. In collaboration with the local team of investigators, the survey company will provide an online training to all the participating interviewers.

## Data collection
### Study 1
Lay people will perform face-to-face interviews in respondents' homes using laptops connected to a secure online survey site (EQ-VT software V.2.1). Interviewers were selected from the members of the national network of hospital assessors managed by the National Authority of Quality Management in Health from Romania and from members of patients' associations. In order to ease respondent recruitment, information about the study will be made available to the general public via social media and, where possible, more specific information will be given to the selected settlements via the Facebook page of the respective city hall (if available) or by contacting key community members.

## Study questionnaire
The survey consists of a computer-assisted interview that uses a version of the EQ-VT software that the EQ foundation developed for the US EQ-5D-5L valuation study.[26] The interview will include the following five-block sections:

1. The EQ-5D-5L questionnaire and background questions on gender, age and experience of illness. EQ-5D-5L, one of the most widely used generic questionnaires to measure HRQoL,[27] is a self-reported questionnaire that asks the respondent to assess his/her current health status on five dimensions (mobility, self-care, usual activities, pain/discomfort and anxiety/depression) and includes a Visual Analogue Scale (EQ-VAS).

2. Composite time trade-off (cTTO) valuations tasks (EQ-5D-3L and 5L). In this section respondents will be asked to imagine living in an impaired health state for a defined period of time and indicate the years of remaining lifetime they will be willing to give up on to avoid living in the impaired health state. Respondents will complete two examples and three practice health states to get used to the task. Then they will assess ten EQ-5D-5L health states and three EQ-5D-3L health states each followed by a feedback module.

3. Discrete choice experiment valuation (DCE) tasks (only for EQ-5D-5L). Respondents will be asked to state their preference between two hypothetical health states. They will value seven pairs of EQ-5D-5L health states, followed by debriefing questions on the valuation tasks.

4. The EQ-5D-3L questionnaire. Respondents will be asked to fill in the EQ-5D-3L questionnaire. EQ-5D-3L measures health status using the same five dimensions as the 5L version, the only difference being that each dimension has only three levels ('no problems', 'some problems' and 'extreme problems').

5. Country-specific questionnaire. The survey will end with a series of sociodemographic questions on residence type, ethnicity, caregiver and parenting status, health literacy (one question that explores the respondent's level of confidence when interacting with healthcare professionals), preference over length or quality of life, marital status, education level, religion (affiliation, general religiosity, participation in religious services, praying), employment status and income.

### Data quality control
The data quality control (QC) process will follow the recommendations of the most up-to-date version of the EQ-VT protocol.[28 29] An online QC tool developed by the EQ Foundation will be used to evaluate interviewers' performance. This tool determines protocol compliance, interviewers' effects and mean values by health state severity[30] and is used by the local team of investigators to provide personalised feedback via phone calls to all interviewers. Data QC checks will be run after each interviewer will have performed a round of 10 interviews. Interviews are flagged as non-compliant if the explanations for the first two examples last for less than 3 min, if the worse than death element is not shown in the examples, if the duration of cTTO tasks is less than 5 min, or if the value given to the worse health state is not the lowest and at least 0.5 higher than that of the state with the lowest value.[30] Poor performing interviewers will be retrained and removed from the team if no improvement is seen after retraining. Checks on the 'random-walk' implementation will also be performed. The local team of investigators will contact by

phone a random selection of respondents (10%) to verify that interviews did take place.

## Study 2

Interviews will be conducted by professional staff from a survey company with experience in working with the Roma community. They will be face-to-face, pen and pencil interviews. Interviewers' access to communities will be facilitated by contacting leaders of the respective communities such as health mediators, social worker assistants or school mediators. All interviews will be conducted with care and sensitivity and respect for all participants' level of literacy. The study will be presented in simple words and if the participant cannot write, the interviewer will sign the informed consent on his/her behalf if he/she desires to take part in the study.

To ease recruitment, just like for study 1, information about the study will be made available to the general public via social media (Facebook page, project webpage) and, if possible, more specific information will be made available to the city halls or health mediators of the selected settlements.

### Study questionnaire

The survey will consist of the following two sections:
1. The EQ-5D-5L questionnaire. The EQ-5D-5L questionnaire was chosen due to its reduced ceiling effects[5] and better distributional parameters and substantial improvement in informativity.[31]
2. Sociodemographic questionnaire. This will include the same questions as the country-specific questionnaire used in study 1 with the addition of two more questions: the availability of health mediators in the respective community and the ability of the respondent to read and write.

### Data QC

The data QC process will be performed by the survey company in accordance with its internal procedures. Paper questionnaires will be validated on a rolling basis and incomplete questionnaires will undergo a 'data recovery process' by recontacting respondents and resuming unanswered items. Additionally, a random selection of 20% of respondents will be contacted by phone to verify that interviews did take place. If needed, field checks will also be performed.

### Analysis
#### Study 1

The EQ-5D-3L value set will be developed using only cTTO data whereas the 5L value set will use both cTTO and DCE data. Different models, such as ordinary least squares, generalised least squares and tobit, will be used to analyse cTTO data, whereas probit or logit models will be used to analyse the DCE data. A hybrid model that uses both cTTO and DCE data will also be developed for the EQ-5D-5L value set using the most innovative modelling techniques.[32 33] The model with the best fit will be chosen following theoretical considerations and comparisons of

models based on the monotonic structure of the decrements, the SEs, the ranking of the dimensions as well as the resulting predictions.

Respondents' responses to both EQ-5D-3L and 5L descriptive systems will be scored using the respective value sets and summary statistics (mean, SD, median, 25th percentile and 75th percentile) by age groups and sex will be presented (population norms for the value set/utility index). The same summary statistics stratified by age groups and sex will also be presented for the EQ-VAS (ie, population norms for EQ-VAS). The distribution of people and the odds ratios for reporting different levels of problems or any problem in all dimensions of the two EQ-5D questionnaires will be calculated and stratified by sociodemographic variables. The frequency of the most common health states in the general population in Romania will also be determined.

The effects of different variables, such as health literacy or religion, on health valuations will also be explored. This will be done using simple and multiple linear regression models that will have as dependent variable the TTO scores. Other explanatory variables considered for these models will be age, sex, place of residence (urban/rural) and educational level. All models will be adjusted for severity of health states with random effects to account for multiple valuations by the same respondent.

#### Study 2

Potential differences in HRQoL according to sociodemographic characteristics and ethnicity will be explored in a subset of age and gender-matched sample of Roma population (study 2) and general population (study 1). If the data follow a normal distribution, t-tests and analysis of variance analysis will be used to reflect differences across groups in HRQoL and for each dimension of the EQ-5D-5L questionnaire, otherwise non-parametric tests will be used. Multiple regression analysis will be used to study how the EQ-5D-5L VAS varies with sociodemographic characteristics, educational level, occupation, income, self-rated health and chronic diseases in the two populations.

For both studies 1 and 2, the survey sample distributions will be adjusted by applying weights based on 2011 census data to the age, sex and regional distribution of the two samples.

### Patient and public involvement

Patients were not involved in the development of the research questions, selection of outcome measures and design of either study 1 or 2 as the approach to address the research questions followed pre-established research methodology. Some of the trained interviewers that performed data collection in study 1 are members of patients' associations in Romania. Suggestions from members of the general public (study 1) and from health mediators and Roma people from Dâmbovița county (study 2) were included in the design of the participant information sheets.

The results of this research will be communicated in two public workshops with local authorities and members of patients' associations and will also be available online at https://research.ncl.ac.uk/qolro/ and at http://www.romaniacurata.ro/valuemed/.

## DISCUSSION

Our study will estimate for the first time in Romania value sets and population norms for the EQ-5D questionnaires. This will constitute a stepping-stone to further development of HTA in Romania as it will potentially lead to more transparent and consistent decision-making in healthcare and to a more efficient use of relatively scarce local resources. It will also provide for the first time a measure of the health status of the general population in Romania and that of Roma communities. This could feed into better public health interventions and policies for either vulnerable or patient groups. Finally, if sample size targets are met, this could become the largest study conducted in Romania on HRQoL.

However, our study has a certain number of limitations. First of all, our respondents will be selected using 'the birthday rule'. 'The birthday rule' is considered to be a quasi-probability procedure of selecting respondents,[34] but it is the easiest and least time-consuming in terms of interviewers' training and administration. It has also been implemented with good results in other representative surveys in Romania and elsewhere.[35]

The survey weights that we will be using to correct the representativeness of the study's samples will be estimated based on the most recent census data available, more exactly the 2011 census data. Since 2011, migration rates have been increasing in Romania with the country currently having the highest growth in the size of its diaspora population after Syria.[36] Unfortunately, the 2011 census data has not yet been updated with the most recent data on the exact number of Romanian emigrants,[37 38] so our weights might not accurately correct the representativeness of the two samples. Also, due to the current troubled political context in Romania, our interviewers might be faced in the field with a lower propensity of local residents to participate in surveys. This could lead to lower than predicted response rates for certain areas, such as urban areas, potentially affecting the quality of the data through non-response bias.

Finally, our survey on Roma communities will consist of self-identified Roma adults. Several studies in Romania have shown that self-identification and hetero-identification generally do not overlap.[39 40] Moreover, self-identified Roma tend to be more impoverished and with a higher residential segregation than hetero-identified Roma.[39] Hence, our comparison results might overestimate the differences between the two groups.

### Ethics and dissemination

As the project involves working with a vulnerable group, the project is of high risk in terms of ethics.

Hence, all interviews will be conducted with care and sensitivity and with respect for participants' ethnicity, religion, language, sexual orientation or literacy level. All participants will be given enough time to read or be read the study's information brochure and to ask questions and discuss concerns regarding potential participation in the study. All the study's materials will have been pretested with volunteers and/or Roma mediators with the aim to try to ensure that information is simple and that words are easy to understand. Participants will be presented the study's participant information sheets, sign the informed consent forms, and be interviewed, all within one visit.

The dissemination plan for this project includes deliverables for the scientific community (scientific articles and conference presentations) and for the general public, such as press releases, brief communications on the project's website, Twitter and Facebook pages. Also, two public workshops with local authorities, physicians and patients' associations will be organised to disseminate the results of the study to a broader audience. Finally, the anonymised datasets with EQ-5D-5L measures will be deposited in a centralised repository, such as Zenodo.

**Acknowledgements** The authors would like to thank Kristina Ludwig and Elly Stolk from the EuroQoL Research Foundation for their comments on the protocol of the general population study (study 1) and support offered during the setup of study 1. We would also like to thank Jan Deckers for his helpful comments and support in preparing the ethical submission for this project. We also thank our interviewers and especially the members of patients' associations in Romania for their help in performing the interviews of study 1.

**Contributors** The study was conceived by IGN-A, EO, YO, LV, EB and SP. EO wrote the first draft of the manuscript. All authors provided their intellectual input and approved the final manuscript.

**Funding** These surveys are conducted by two teams of researchers that applied independently for funding. Given that several support activities are common across surveys, the two teams decided to offer each other methodological and scientific support. Hence, these surveys have several funding sources. The estimation of EQ-5D-5L value set, population norms and the comparison with Roma communities received funding from the European Union's Horizon 2020 research and innovation programme under the Marie Skłodowska-Curie grant agreement No 748612. It has also received funding from the EuroQol Research Foundation. The estimation of EQ-5D-3L value set and population norms received partial funding as part of a project run through the Romanian Operational Programme 'Administrative Capacity' (POCA 2014-2020).

**Map disclaimer** The depiction of boundaries on the map(s) in this article do not imply the expression of any opinion whatsoever on the part of BMJ (or any member of its group) concerning the legal status of any country, territory, jurisdiction or area or of its authorities. The map(s) are provided without any warranty of any kind, either express or implied.

**Competing interests** None declared.

**Patient consent for publication** Not required.

**Ethics approval** This study will be conducted in agreement with ethical principles set out in the Declaration of Helsinki and Regulation (EU) 679/2016 on the protection of natural persons with regard to the processing of personal data and the free movement of such data. Ethical approval was sought from the Bioethics Committee of Medicines and Medical Devices, Romania and from the Faculty of Medical Sciences Research Ethics Committee, part of Newcastle University''s Research Ethics Committee, UK.

**Provenance and peer review** Not commissioned; externally peer reviewed.

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
