## [Reviewer comments · BMJ Open]

ARTICLE DETAILS

TITLE (PROVISIONAL)	Measuring health-related quality of life in the general population and Roma communities in Romania: study protocol for two cross-sectional studies
AUTHORS	Olariu, Elena; Paveliu, Sorin; Baican, Eugen; Oluboyede, Yemi; Vale, Luke; NICULESCU-ARON, ILEANA GABRIELA

VERSION 1 – REVIEW

REVIEWER	Asrul Akmal Shafie Discipline of Social and Administrative Pharmacy School of Pharmaceutical Sciences Universiti Sains Malaysia Malaysia
REVIEW RETURNED	12-Apr-2019

GENERAL COMMENTS	Spelling error : Page 6 of 19 : Line 27 & 30 : Omibus survey = Omnibus survey Comments : The 5 blocks of questionnaires for Study One is pretty heavy on the interviewer & respondent. Wonder if it might cause interviewer/respondent fatigue and affect the outcome? Can the EQ-VT Software capture the quality accurately in this situation? Was it meant for 3L & 5L valuation simultaneously? Study One: Part 1: Developing Romanian Value Sets EQ-5D-3L & 5L 1. Data Quality Control : Utilizes online QC tool by EuroQol that automates production of QC reports but criteria of 'flagging' a poor performing interviewer is not defined. 2. Statistical Analysis : New methods of modelling EQ-5D-5L (Feng Y, Devlin NJ, Shah KK, Mulhern B, van Hout B. New methods for modelling EQ-5D-5L value sets: An application to English data. Health Economics. 2018;27:23–38.) Just adding this on for alternative methods to try maybe (?) since it wasn't in the list of references. Study One: Part 2: Population Norms for Romanian General Population using EQ-5D-3L & 5L 1. Statistical analysis : Not described in depth. How will the population norm be determined? Using mean values? Study Two: Population Norms for Roma Population To check normality of HRQoL before using t-test/ANOVA. Most HRQoL data will not have a normal distribution
--

REVIEWER	Juanita Haagsma Erasmus MC University Medical Center, The Netherlands
REVIEW RETURNED	12-Apr-2019

GENERAL COMMENTS	This protocol describes the study aims, sampling strategy and study design of each of the sub studies very clearly. A remark: In the introduction section the authors write: "Health-Related Quality of Life (HRQL) can be described as a patient's emotional view on the effect illness and its associated treatment(s) can have on his/her everyday life, physical, mental, social functioning and prosperity." Can we describe it as a patient's emotional view? Do you from patient's perspective? Emotional view and patient's perspective are not the same concepts. Third stage, line number 28: "he/she will start with the household located at the lowest household number on the street". Do you means lowest house number in stead of household number on the street? If not, I do not understand what you mean by this and suggest to rephrase. The study questionnaire of study 1 includes the EQ-5D-3L, whereas the questionnaire of study 2 includes the EQ-5D-5L and EQ-5D-3L questionnaire. Why do you include both versions of the questionnaire of study 1? Why not only include the EQ-5D-5L?
--

VERSION 1 – AUTHOR RESPONSE

Reviewer(s)' Comments to Author: Reviewer: 1; Reviewer Name: Asrul Akmal Shafie

Institution and Country: Discipline of Social and Administrative Pharmacy School of Pharmaceutical Sciences, Universiti Sains Malaysia

Please state any competing interests or state 'None declared': None declared

Please leave your comments for the authors below

Spelling error :

Page 6 of 19 : Line 27 & 30 : Omibus survey = Omnibus survey

Authors' response:

We have now corrected and checked again our manuscript for spelling mistakes.

Methods and analysis

I. Aim and objectives

Our specific objectives are:

1. To develop a value set for EQ-5D-3L and 5L based on societal preferences in Romania (Study one, an Omnibus survey)
2. To determine the population norms of the Romanian version of EQ-5D-3L and 5L (Study one, an Omnibus survey)

Comment 1

The 5 blocks of questionnaires for Study One is pretty heavy on the interviewer & respondent. Wonder if it might cause interviewer/respondent fatigue and affect the outcome? Can the EQ-VT Software capture the quality accurately in this situation? Was it meant for 3L & 5L valuation simultaneously?

Authors' response:

Thank you very much for your comment: the burden of the interview, and specifically of the time-trade off tasks (TTO), is indeed high for both the interviewer and respondent. For the 301 interviews we have conducted up to the 31st of March 2019 the average duration of an interview was of 49 minutes (SD=19 minutes) and the average duration of a TTO task was of 1.35 minutes (SD=1.1 minutes). In order to reduce the burden, face-to-face interviewing was chosen to administer the questionnaires in spite of the higher costs. This mode conveys more easily interviewers' engagement and enthusiasm to respondents, making them more motivated and less likely to shortcut the tasks (1). Additionally, the QC tool developed by the EuroQoL Foundation can be used to detect respondents' satisficing by observing and comparing the TTO value distribution by interviewer with the overall TTO value distribution. If satisficing is observed, interviewers are advised, by phone or email, to engage respondents more either by providing additional explanations in words easier to understand, or by encouraging respondents to think-aloud or by reminding them the importance of the study. In terms of interviewers' fatigue, our interviewers are hired part-time, so they generally perform less than five interviews per day. Additionally, interviews are performed in respondents' homes thus helping to reduce the potential boredom associated with repeating under the same circumstances the same scenario multiple times.

The EQVT software was developed to allow the estimation of the EQ-5D-5L value set. However, the software does allow the addition of other methodological objectives. The version of the software we are currently using was initially developed for the US EQ-5D-5L valuation study (2). This valuation study included an additional sub-aim to directly compare EQ-5D-3L and 5L value sets developed using time trade-off (TTO) values. After the US valuation study, the same software was used in Hungary to develop both 3L and 5L value sets (study still ongoing). Given that Romania does not currently have a value set for either of the EQ-5D versions, we decided to use this software to simultaneously develop the 3L and 5L value sets for our country as it allows an improved quality control.

A. Study One: Part 1: Developing Romanian Value Sets EQ-5D-3L & 5L

Comment A.1 - Data Quality Control: Utilizes online QC tool by EuroQoL that automates production of QC reports but criteria of 'flagging' a poor performing interviewer is not defined.

Authors' response:

According to the quality control process developed by the EuroQoL foundation, an interview is flagged if either the worse than death element is not shown in the wheelchair example, or the duration of the explanations provided for the cTTO task in the wheelchair example is less than three minutes, or if the duration of the ten cTTO tasks for EQ-5D-5L is less than five minutes or if the value for the worse health state (55555) is not the lowest or at least 0.5 higher than the value assigned to the health state with the lowest value (3).

We have now included a short description of the criteria for flagging a poor performing interviewer in the manuscript as well.

Methods and analysis

V. Data collection

Study One

B. Data quality control

Data quality control checks will be run after each interviewer will have performed a round of 10 interviews. Interviews are flagged as non-compliant if the explanations for the first two examples last for less than three minutes, if the worse than death (WTD) element is not shown in the examples, if the duration of cTTO tasks is less than five minutes, or if the value for the worse health state is not the lowest and at least 0.5 higher than that of the state with the lowest value³⁰. Poor performing interviewers will be re-trained and removed from the team if no improvement is seen after re-training.

Comment A.2 - Statistical Analysis: New methods of modelling EQ-5D-5L (Feng Y, Devlin NJ, Shah KK, Mulhern B, van Hout B. New methods for modelling EQ-5D-5L value sets: An application to English data. *Health Economics*. 2018;27:23–38.) Just adding this on for alternative methods to try maybe (?) since it wasn't in the list of references.

Authors' response:

Thank you for the comment: we have now updated the Analysis section of our manuscript.

Methods and analysis

VI. Analysis

Study one

A hybrid model that uses both cTTO and DCE data will also be developed for the EQ-5D-5L value set using the most innovative modelling techniques^{31 32}.

B. Study One: Part 2: Population Norms for Romanian General Population using EQ-5D-3L & 5L

Comment B.1 - Statistical analysis: Not described in depth. How will the population norm be determined? Using mean values?

Authors' response:

EQ-5D-3L & 5L responses will be scored using the value sets developed. Summary statistics, such as mean, standard deviation, median, percentile 25 and 75 will be presented by age groups and sex. We will also present summary statistics (mean, standard deviation, median, percentile 25 and 75) for the EQ visual analogue scale (EQ-VAS).

The Analysis section of our manuscript has now been updated to include this information.

Methods and analysis

VI. Analysis

Study one

Respondents' responses to both EQ-5D-3L and 5L descriptive systems will be scored using the respective value sets and summary statistics (mean, standard deviation, median, 25th percentile and 75th percentile) by age groups and sex will be presented (population norms for the value set/utility index). The same summary statistics stratified by age groups and sex will also be presented for the EQ-VAS (population norms for EQ-VAS). The distribution of people and the odds ratios for reporting different levels of problems or any problem in all dimensions of the two EQ-5D questionnaires will be calculated and stratified by sociodemographic variables. The frequency of the most common health states in the general population in Romania will also be determined.

C. Study Two: Population Norms for Roma Population

Comment C.1 - To check normality of HRQoL before using t-test/ANOVA. Most HRQoL data will not have a normal distribution.

Authors' response:

Thank you for the comment: we have now updated the Analysis section of Study two.

Methods and analysis

VI. Analysis

Study two

If the data follows a normal distribution, t-tests and ANOVA analysis will be used to reflect differences across groups in HRQoL and for each dimension of the EQ-5D-5L questionnaire, otherwise non-parametric tests will be used.

Reviewer: 2

Reviewer Name: Juanita Haagsma

Institution and Country: Erasmus MC University Medical Center, The Netherlands

Please state any competing interests or state 'None declared': None declared

Please leave your comments for the authors below

This protocol describes the study aims, sampling strategy and study design of each of the sub studies very clearly.

Comment 1

In the introduction section the authors write: „Health-Related Quality of Life (HRQL) can be described as a patient's emotional view on the effect illness and its associated treatment(s) can have on his/her everyday life, physical, mental, social functioning and prosperity.” Can we describe it as a patient's emotional view? Do you (measure it?) from patient's perspective? Emotional view and patient's perspective are not the same concepts.

Authors' response:

Thank you for the comment. We agree with the reviewer that defining health-related quality of life (HRQoL) is a challenging task with multiple definitions being reported in the literature (4). HRQoL is a multidimensional concept that includes both objective and subjective data (5, 6), with the majority of HRQoL measures capturing self-perceived health status (4). The gold standard in measuring HRQoL are self-reported measures, with proxies being used only when a person is unable or too young to report it themselves.

We have now updated the Introduction section of our manuscript.

Introduction

HRQoL is a multidimensional concept that includes physical, psychological, functional, and social aspects¹ regarding a person's perception of the impact of health status on quality of life².

Comment 2

Third stage, line number 28: „he/she will start with the household located at the lowest household number on the street“. Do you means lowest house number instead of household number on the street? If not, I do not understand what you mean by this and suggest to rephrase.

Authors' response:

The reviewer's interpretation is correct: the first household selected in the random walk is the one found at the lowest house number on the street. We have now updated the corresponding text in the manuscript.

Methods and analysis

III. Study population

C. Sample design

Study one

Once the street has been identified, he/she will start with the household located at the lowest house number on the street.

Comment 3

The study questionnaire of study 1 includes the EQ-5D-3L, whereas the questionnaire of study 2 includes the EQ-5D-5L and EQ-5D-3L questionnaire. Why do you include both versions of the questionnaire of study 1? Why not only include the EQ-5D-5L?

Authors' response:

Study one includes both EQ-5D-3L and EQ-5D-5L as it aims to develop value sets and population norms for both questionnaires given that in Romania neither are available. Study two aims to compare the health-related quality of life of the Roma communities with that of the majority of the population in Romania. In study two health-related quality of life is measured only with EQ-5D-5L. We chose EQ-5D-5L due to its arguably improved properties compared with the 3L variant, such as more reduced ceiling effects (7) and better distributional parameters and substantial improvement in informativity (8).